

# Factors associated with incomplete latent tuberculosis infection preventive treatment in Sabah, Malaysia

Ahmad Firdaus Mohamed[1], Mohd Nazri Shafei[1], Wan Mohd Zahiruddin Wan Mohammad[1], Roddy Teo[2] and Amabel Seow Min Hui[2]

[1] Department of Community Medicine, School of Medical Sciences, Universiti Sains Malaysia, Kota Bharu, Kelantan, Malaysia
[2] Disease Control Division (TB and Leprosy Sector), Sabah State Health Department, Kota Kinabalu, Sabah, Malaysia

## ABSTRACT

**Background.** Latent tuberculosis infection (LTBI) is a critical public health issue in Malaysia, particularly in regions like Sabah, where the incidence of tuberculosis (TB) remains high. LTBI can progress to active TB if left untreated, making preventive treatment essential in reducing TB transmission. However, adherence to LTBI preventive treatment remains a significant challenge, with incomplete treatment potentially undermining efforts to control TB. This study aimed to determine the proportion of individuals with LTBI who did not complete preventive treatment and to identify associated factors.

**Methods.** A retrospective record review was conducted among individuals with LTBI registered in the Sabah State Health Department's LTBIS 401A registry. Multiple logistic regression analyses were applied to determine the factors associated with incomplete preventive treatment.

**Results.** A total of 895 individuals with LTBI were included in the study. The proportion of incomplete LTBI preventive treatment was 9.2%. Factors that were significantly associated with the incomplete preventive treatment were non-HCW occupation (adj.OR = 4.21, 95 CI [1.25–14.22]), residents of Tawau Division (adj.OR = 2.00, 95% CI [1.10–3.65]), and individuals with LTBI without contact to TB patients (adj.OR = 2.79, 95% CI [1.42–5.48]).

**Conclusion.** The proportion of incomplete preventive treatment among individuals with LTBI in Sabah was comparatively lower than many previous studies. Targeted interventions should be developed to address the specific needs of the groups with higher odds of having incomplete preventive treatment. It includes tackling the social determinants of health, like improving healthcare system accessibility. A prospective study to evaluate these interventions' effectiveness in improving preventive treatment completion rate is recommended.

Corresponding author
Mohd Nazri Shafei, drnazri@usm.my

## INTRODUCTION

In 2014, an estimated 23% of the world's population, or roughly 1.7 billion people, were infected with latent tuberculosis. The highest rates of latent tuberculosis infection (LTBI) were found in Southeast Asia, the Western Pacific, and Africa, which together accounted for around 80% of all LTBI cases (*Houben & Dodd, 2016*). Malaysia is an upper-middle-income country with an upper-moderate tuberculosis (TB) disease burden (*The World Bank Group, 2022*; *World Health Organization (WHO), 2021*). In Malaysia, the burden of TB in 2022 was significant, with 25,391 reported cases of active TB disease (incidence rate of 77.8 per 100,000) and 2,572 TB-related deaths (*Monihuldin, 2023*). However, as of now, there is no published data on LTBI prevalence among general population in Malaysia (*Romli & Sabirin, 2020*).

To achieve the long-term goal of TB elimination, the World Health Organization (WHO) is pushing for more comprehensive preventive treatment of LTBI (*United Nations, 2022*; *World Health Organization (WHO), 2020*). In Malaysia, the Ministry of Health developed a pilot project in 2019 that utilised interferon-gamma release assays (IGRA) and tuberculin skin test (TST) to detect and manage LTBI. The pilot project was subsequently implemented nationwide in August 2020. In Malaysia, the first line LTBI treatment recommendation is daily Isoniazid and Rifampin for three months (3HR) or daily Isoniazid and Rifapentine for three months (3HP), however the drugs availability are still limited (*Ministry of Health Malaysia, 2021*). The latent Tuberculosis Information System (LTBIS) registry was established in 2020 following the implementation of programmatic screening of LTBI individuals (*Ministry of Health Malaysia, 2020b*).

Sabah, one of 13 states of Malaysia persistently recorded the highest cases of TB in Malaysia, with more than 5,238 cases (incidence rate of 134.2 per 100,000) reported in 2019 (*Department of Statistics Malaysia, 2022*). The state is in East Malaysia and is economically less developed than the other states in West Malaysia. The native ethnicities in Sabah differ from those in West Malaysia. Sabah also has many legal and illegal immigrants from the Philippines and Indonesia (*Avoi & Liaw, 2021*; *Dollah et al., 2016*). The Philippines and Indonesia are the top 30 TB burden countries, with an incidence rate of 539 and 301 per 100,000 population, respectively (*World Health Organization (WHO), 2021*; *World Health Organization (WHO), 2022*). The state of Sabah persistently recorded the highest case of TB in Malaysia, with more than 5,238 cases (incidence rate of 134.2 per 100,000) reported in 2019 (*Department of Statistics Malaysia, 2022*). Furthermore, immigrants contributed more than 24% of new cases detected since 1990 (*Dony, Ahmad & Khen Tiong, 2004*). The TB mortality rate in Sabah alarmingly increases yearly (*Avoi & Liaw, 2021*).

The incomplete preventive treatment of LTBI is concerning as it can result in the progression of the infection to active TB. There has not been any previous research on the completion of LTBI preventive treatment in Malaysia, as the nationwide access to IGRA test and programmatic screening of LTBI are relatively recent developments. Hence, this study was designed to determine the proportion and factors associated with

incomplete LTBI preventive treatment in Sabah, Malaysia with the ultimate goal of providing guidance to policymakers on LTBI management.

## MATERIALS & METHODS

### Study design and location

The study used a retrospective record review with a cohort design. The data were obtained from the Latent TB Information System (LTBIS 401A) registry of Sabah State Health Department after obtaining permission from the institution. Given the retrospective nature of the study, informed consent, either verbal or written, was not required from the study participants. Sabah is located at the north of Borneo Island and geographically separated from West Malaysia by the South China Sea. Internationally, Sabah shares borders with the Philippines and Indonesia. Sabah is the second largest state in Malaysia with land area of 73,904 square kilometers. About 3.9 million people live in Sabah, with almost 30% being immigrants. Sabah comprises of five administrative divisions, which are further divided into 27 districts (*Department of Statistics Malaysia, 2023*). The five divisions are West Coast, Tawau, Sandakan, Kudat, and Interior Division. In 2019, the state median monthly household income was RM 4235, much lower than the national median income of RM 5873 (*Department of Statistics Malaysia, 2020a*).

### Study population

Using a single proportion formula with proportion of incomplete preventive treatment of 28% (*Kawatsu, Uchimura & Ohkado, 2017*) and the precision set at 3%, the required sample size for the study was 957 after considering 10% incomplete data. We included individuals with LTBI registered in the LTBIS 401A registry in Sabah between 1st January 2019 and 30th July 2022. We excluded those who developed active TB during treatment, those who died for any reason during treatment, and those who were transferred to other treatment centers. The cohort of individuals were followed up monthly until the completion of the preventive treatment.

Those who were tested positive for either IGRA or TST or both, and who did not have TB lesions on chest X-ray, as well as no clinical TB symptoms or signs was considered as having latent TB infection (*Ministry of Health Malaysia, 2020b*). Any individual who had an interruption of treatment for LTBI for more than or equal to two months or loss of follow-up for more than or equal to two months was considered as having incomplete LTBI preventive treatment (*Ministry of Health Malaysia, 2020b*). Individuals with any of the following conditions: diabetes mellitus, HIV, chronic renal failure, organ transplant, or cancer, were defined as having comorbidities.

Sabah ethnicity includes all ethnicities that originate from Sabah state, including Kadazan, Dusun, Murut, Bajau, and others. Non-Sabah ethnicity consists of all ethnicities that originate outside of Sabah, such as Malay, Chinese, Indian, and Iban. Non-Malaysian ethnicities include Indonesian and Filipino. Unknown ethnicity refers to individuals with missing ethnicity status that cannot be further verified.

In Malaysia, LTBI preventive treatment is provided free of charge through public healthcare facilities as part of the National TB Control Programme. Once an individual

is diagnosed with LTBI and initiated on treatment, healthcare workers (HCWs) at TB clinics conduct monthly follow-up visits to monitor treatment adherence, assess for side effects, and provide counseling. If an individual misses a scheduled appointment, TB clinic staff—together with health inspectors from the District Health Office—initiate follow-up *via* phone calls to re-engage the individual in care. However, Directly Observed Therapy (DOT) is not routinely implemented for LTBI cases. Adherence is primarily self-managed and monitored through clinic visits.

Importantly, under Malaysia's Prevention and Control of Infectious Diseases Act (CDC Act), non-completion of LTBI preventive treatment is not considered a legal offense. Therefore, adherence to LTBI treatment relies entirely on individual willingness, motivation, and perception of the treatment's benefits, without any legal enforcement mechanism. This may particularly affect individuals facing structural barriers or those with limited understanding of LTBI risks.

### Research tool

A proforma was utilised to extract anonymized and pertinent data solely from LTBIS 401A for this study. The data were then inputted into Microsoft Excel. The variables that were extracted onto the proforma were age, gender, ethnicity, nationality, occupational sector, residential region, contact to TB case status, CRF on dialysis status, HIV/AIDS status, cancer status, steroid treatment status, diabetes mellitus status, organ transplant status, treatment regime, medication adverse reaction and completion of preventive treatment status as the outcome.

### Data analyses

Data were analysed using SPSS version 26 (IBM, Armonk, USA). The sociodemographic characteristics of the subjects were analysed using descriptive statistics. Based on their normality distribution, the numerical data were displayed as mean (standard deviation) or median (interquartile range). Categorical data were expressed as frequency (percentage). The denominator used for the analysis was the individuals with LTBI in Sabah who fulfilled the study criteria.

To assess the factors associated with incomplete preventive treatment, the statistical analysis method used was multiple logistic regression. Preventive treatment completion status was divided into complete and incomplete preventive treatment groups. The outcome was a binary variable coded "0" for complete preventive treatment and "1" for incomplete preventive treatment.

Initially, simple logistic regression was conducted to assess the effect of each independent variable, and the results were presented as crude odds ratio (OR). Subsequently, multiple logistic regression was performed using forward and backward selection methods. The variable selection included only variables with a *p*-value of less than 0.25 (*Sperandei, 2014*). Further, multicollinearity and interaction among the significant variables were also examined. Subsequently, the model's goodness of fit was evaluated using several tests, including the Hosmer and Lemeshow test, classification table, and area under the receiver operating characteristic curve tests (*Chowdhury & Turin, 2020*; *Sperandei, 2014*).

 

## Ethical statement

The study was approved by the Malaysian Medical Research and Ethics Committee (NMRR ID-22-02792-6PV (IIR)) and the Human Research Ethics Committee of Universiti Sains Malaysia (USM/JEPeM/22110728). To maintain data confidentiality, participant identities were anonymized, and all data were securely stored with limited access granted solely to authorized personnel. Subsequently, all remaining data was coded to provide additional protection for participant confidentiality.

## RESULTS

There were 916 cases of LTBI registered between 1st January 2019 and 30th July 2022 in the LTBIS 401A registry. All individuals who fulfilled the subject criteria from the sampling frame were included in the study. Twenty-one individuals had missing data that could not be further verified and were excluded from the study. As a result, the study included 895 individuals. Table 1 presents the sociodemographic and clinical characteristics of individuals with completed and incomplete preventive treatment. In this study, 82 individuals (9.2%) with LTBI did not complete the preventive treatment. Regarding comorbidities, none of the individuals included in the study were reported to have chronic renal failure, cancer, or were undergoing steroid treatment. Two individuals were identified as HIV reactive; of these, one completed, and one did not complete LTBI preventive treatment. The individual with incomplete treatment had experienced a medication adverse reaction. However, no additional clinical or programmatic details were available in the registry to fully contextualize the treatment discontinuation.

In Table 2, all the individuals with LTBI with incomplete preventive treatment were further analyzed by year of registration. Of all individuals with incomplete preventive treatment, the highest proportion of incomplete preventive treatment was in 2019 (27.9%).

In Table 3, the multiple logistic regression analysis revealed that occupation, residential region, and TB contact status are significantly associated with incomplete preventive treatment. Being non-HCW has 4.21 times higher odds of having incomplete preventive treatment (95% CI [1.25–14.22], $p$-value = 0.021) compared to HCW when adjusted for the residential region and TB contact status. Staying in Tawau Division has 2.00 times higher odds of having incomplete preventive treatment (95% CI [1.10–3.65], $p$-value = 0.023) compared to the West Coast Region when adjusted for the occupation and TB contact status. Non-TB contact has 2.79 times higher odds of incomplete preventive treatment (95% CI [1.42–5.48], $p$-value = 0.003) compared to TB contact when adjusted for the occupation and residential region.

## DISCUSSION

In 2019, the LTBI screening and treatment program began with 43 registered individuals in the LTBIS 401A registry. However, in 2020, the COVID-19 pandemic led to a significant decline in registrations, with only 27 individuals due to government measures like

**Table 1  Characteristics of individuals with completed and incomplete LTBI preventive treatment in Sabah (*n* = 895).**

| Variables | | *n* (%) | Incomplete preventive treatment (*n* = 82) *n* (%) | Complete preventive treatment (*n* = 813) *n* (%) |
|---|---|---|---|---|
| **Gender** | | | | |
| | Male | 357 (39.9) | 36 (10.1) | 321 (89.9) |
| | Female | 538 (60.1) | 46 (8.6) | 492 (91.4) |
| **Age group** | | | | |
| | 0–24 | 250 (27.9) | 17 (6.8) | 233 (93.2) |
| | 25–44 | 324 (36.2) | 29 (9.0) | 295 (91.0) |
| | 45–64 | 268 (29.9) | 32 (11.9) | 236 (88.1) |
| | ≥65 | 53 (5.9) | 4 (7.5) | 49 (92.5) |
| **Ethnicity** | | | | |
| | Sabah ethnicity | 613 (68.5) | 48 (7.8) | 565 (92.2) |
| | Non-Sabah ethnicity | 53 (5.9) | 8 (15.1) | 45 (84.9) |
| | Non-Malaysian | 41 (4.6) | 9 (22.0) | 32 (78.0) |
| | Unknown | 188 (21) | 17 (9.0) | 171 (91.0) |
| **Nationality** | | | | |
| | Malaysian | 854 (95.4) | 73 (8.5) | 781 (91.5) |
| | Immigrant | 41 (4.6) | 9 (22.0) | 32 (78.0) |
| **Occupation** | | | | |
| | HCW[a] | 100 (11.2) | 3 (3.0) | 97 (97.0) |
| | Others | 795 (88.8) | 79 (9.9) | 716 (90.1) |
| **Residential region** | | | | |
| | West Coast division | 343 (38.3) | 24 (7.0) | 319 (93.0) |
| | Tawau division | 176 (19.7) | 26 (14.8) | 150 (85.2) |
| | Sandakan division | 198 (22.1) | 22 (11.1) | 176 (88.9) |
| | Kudat division | 47 (5.3) | 4 (8.5) | 43 (91.5) |
| | Interior division | 131 (14.6) | 6 (4.6) | 125 (95.4) |
| **Contact to TB case** | | | | |
| | Yes | 813 (90.8) | 68 (8.4) | 745 (91.6) |
| | No | 82 (9.2) | 14 (17.1) | 68 (82.9) |
| **Comorbidities** | | | | |
| | Yes | 18 (2.0) | 1 (5.6) | 17 (94.4) |
| | No | 877 (98.0) | 81 (9.2) | 796 (90.8) |
| **Treatment regime** | | | | |
| | 6H[b] | 877 (98.0) | 80 (9.1) | 797 (90.0) |
| | Others | 18 (2.0) | 2 (11.1) | 16 (88.9) |
| **Medication adverse reaction** | | | | |
| | Yes | 24 (2.7) | 24 (100.0) | 0 (0.0) |
| | No | 871 (97.3) | 58 (6.7) | 813 (93.3) |

Notes.
[a] Healthcare worker.
[b] Six months of daily isoniazid.

**Table 2** Proportion of incomplete preventive treatment among LTBI patients in Sabah ($n = 895$).

| Year | Total LTBI patients | Incomplete preventive treatment $n$ (%) | Completed preventive treatment $n$ (%) |
|------|------|------|------|
| 2019 | 43 | 12 (27.9) | 31 (72.1) |
| 2020 | 27 | 3 (11.1) | 24 (88.9) |
| 2021 | 478 | 46 (9.6) | 432 (90.4) |
| 2022 | 347 | 21 (6.1) | 326 (93.9) |

the Movement Control Order (MCO). As the MCO was eased, registrations increased, reaching 478 in 2021.

The study found that only 9.2% of individuals with LTBI had incomplete preventive treatment, which is relatively low compared to global rates ranging from 9% in Norway to 53.8% in the US (*Schein et al., 2018*; *Stockbridge et al., 2018*). The variation in incomplete preventive treatment rates can be linked to the healthcare financing system. Countries with compulsory public taxation or social health insurance systems tend to have lower rates than those with private insurance or out-of-pocket payments. Malaysia's public tax-funded healthcare system, provides free LTBI treatment to both citizens and immigrants (*Yu, Whynes & Sach, 2008*). Effective case management, contact tracing, and defaulter tracing by primary healthcare facilities also play a role.

This study in Sabah revealed a significant association between non-healthcare workers (non-HCW) and incomplete LTBI preventive treatment. This finding contrasts with other studies that found HCWs were significantly associated with incomplete LTBI preventive treatment (*Horsburgh Jr et al., 2010*; *Kawatsu, Uchimura & Ohkado, 2017*). Non-HCWs may encounter greater challenges in accessing LTBI preventive treatment compared to HCWs. Barriers such as transportation issues and geographic distance to healthcare facilities could impede non-HCWs' ability to complete LTBI preventive treatment regimens. Although treatment itself is free, the indirect costs incurred for transportation can pose significant financial burdens. Moreover, the operating hours of healthcare facilities may not align with non-HCWs' work schedules, making it difficult for them to attend follow-up appointments without risking their livelihoods. Conversely, HCWs may enjoy easier access to healthcare services and resources, which could facilitate their completion of LTBI preventive treatment. Indirect cost such as transport and skipping work need to be considered to ensure equitable access to TB prevention and control measures.

While non-HCWs in this study had significantly higher odds of incomplete treatment, it is noteworthy that 3% of HCWs also did not complete LTBI preventive therapy. This finding aligns with previous literature showing that even medically trained individuals are not immune to adherence challenges. A systematic review by *Yang & Park (2022)* highlighted that common reasons for non-completion among HCWs include adverse drug reactions, low perceived risk of disease progression, and logistical barriers such as workload or scheduling conflicts. These findings suggest that treatment adherence

**Table 3** Associated factors of incomplete preventive treatment among LTBI patients in Sabah, using simple and multiple logistic regression ($n = 895$).

| Variables | Crude OR (95% CI) | Wald stat (df) | p-value | Adj. OR (95% CI) | Wald stat (df) | p-value |
|---|---|---|---|---|---|---|
| **Gender** | | | | | | |
| Female | 1 | | | | | |
| Male | 1.20 (0.76, 1.90) | 0.61 (1) | 0.437 | | | |
| **Age group (years)** | | | | | | |
| 0–24 | 1 | | | | | |
| 25–44 | 1.35 (0.72, 2.51) | 3.90 (1) | 0.348 | | | |
| 45–64 | 1.86 (1.00, 3.44) | 0.04 (1) | 0.048 | | | |
| ≥ 65 | 1.12 (0.36, 3.47) | 0.88 (1) | 0.846 | | | |
| **Nationality** | | | | | | |
| Malaysian | 1 | | | | | |
| Immigrant | 3.01 (1.38, 6.55) | 7.71 (1) | 0.005 | | | |
| **Ethnicity** | | | | | | |
| Sabahan | 1 | | | | | |
| Non-sabahan | 2.09 (0.93, 4.69) | 3.21 (1) | 0.073 | | | |
| Non-Malaysian | 3.31 (1.49, 7.34) | 8.69 (1) | 0.003 | | | |
| Unknown | 1.17 (0.66, 2.09) | 0.28 (1) | 0.595 | | | |
| **Comorbidities** | | | | | | |
| No | 1 | | | | | |
| Yes | 0.58 (0.08, 4.40) | 0.28 (1) | 0.597 | | | |
| **Treatment Regime** | | | | | | |
| 6H[c] | 1 | | | | | |
| Others | 1.25 (0.28, 5.51) | 0.08 (1) | 0.773 | | | |
| **Occupation** | | | | | | |
| HCW[a] | 1 | | | 1 | | |
| Non-HCW | 3.57 (1.11, 11.52) | 4.52 (1) | 0.033 | 4.21 (1.25, 14.22) | 5.35 (1) | 0.021 |
| **Residential region** | | | | | | |
| West Coast Division | 1 | | | 1 | | |
| Tawau Division | 2.30 (1.28, 4.15) | 7.75 (1) | 0.005 | 2.00 (1.10, 3.65) | 5.17 (1) | 0.023 |
| Sandakan Division | 1.66 (0.91, 3.05) | 2.69 (1) | 0.101 | 1.61 (0.87, 2.99) | 2.27 (1) | 0.132 |
| Kudat Division | 1.24 (0.41, 3.73) | 0.14 (1) | 0.707 | 1.24 (0.41, 3.80) | 0.15 (1) | 0.703 |
| Interior Division | 0.64 (0.26, 1.60) | 0.92 (1) | 0.337 | 0.69 (0.27, 1.74) | 0.63 (1) | 0.426 |
| **Contact to TB case** | | | | | | |
| Yes | 1 | | | 1 | | |
| No | 2.26 (1.21, 4.22) | 6.48 (1) | 0.011 | 2.79 (1.42, 5.48) | 8.82 (1) | 0.003 |

**Notes.**
[a] Healthcare worker.
No multicollinearity and interaction.
Hosmer-Lemeshow test, p-value = 0.993.
Classification table = 90.8%.
Area under the curve = 64.4%.

among HCWs is influenced not only by clinical factors but also by personal beliefs and occupational demands. Tailored strategies—such as adherence counseling, flexible follow-up systems, and the use of shorter, less toxic regimens—may be beneficial in supporting HCWs through treatment completion.

This study identified a significant association between incomplete LTBI preventive treatment and the region of residence. Individuals with LTBI in the Tawau Division had significantly higher odds (2.00 times) of incomplete LTBI preventive treatment compared to the West Coast Division, which was used as a reference. This finding can be understood through the social determinants of health framework, which play a crucial role to determine health outcomes (*Huang, Morgan & Yoshino, 2019*; *World Health Organization (WHO), 2003*; *World Health Organization (WHO), 2010*; *Wilkinson & Pickett, 2011*). In the case of the Tawau division, the absence of accessible and safe transportation, particularly in a region with many islands, significantly hinders individuals with LTBI from continuing and completing their preventive treatment. The high wealth inequality in Tawau, with a considerable population living below the poverty line alongside significant income disparities, exacerbates the challenges. The districts in the Tawau Division have among the highest wealth inequality, with the poor being really poor and the rich being really well-off (*Department of Statistics Malaysia, 2020b*).

Furthermore, based on the human resource report by the Ministry of Health, Sabah had the lowest density of HCW. There were only 11.73 doctors per 10,000 population compared to the national average of 18.88 doctors per 10,000 population (*Ministry of Health Malaysia, 2020a*). This results in reduced healthcare system accessibility, further hindering individuals from completing the extended duration of preventive treatment. Given the presence of these multiple social determinants of health in the Tawau Division, the significant association between this region and incomplete LTBI preventive treatment is not surprising.

Among the individuals with LTBI included in the study, only 82 (9.2%) individuals were non-contact to TB patients. The individuals with LTBI who were non-contact to TB patients had 2.79 (95% CI [1.42–5.48]) higher odds of incomplete LTBI preventive treatment compared to individuals with LTBI with contact to TB patients. This is aligned with a study by *Shieh et al. (2006)* in the US, which found that individuals who perceived a low risk for developing active TB without LTBI treatment were significantly less likely to complete LTBI treatment (RR: 0.35, 95% CI [0.18–0.67]). Using the Health Belief Model (HBM) as a framework, it can be understood why non-contact with TB patients is associated with incomplete LTBI preventive treatment. Individuals who didn't have direct contact with TB patients may perceive a lower risk of acquiring LTBI, leading to a reduced perceived need for preventive treatment. Consequently, they may underestimate the benefits of LTBI preventive treatment. Furthermore, individuals without contact with TB patients may not have the same cues for action to continue LTBI preventive treatment.

In the univariate analysis, both immigrant status and non-Sabahan or non-Malaysian ethnicity were significantly associated with higher odds of incomplete LTBI preventive treatment. However, these associations did not remain statistically significant in the multiple logistic regression model after adjusting for occupation, region, and TB contact

status. This suggests potential confounding by these covariates. Despite the lack of independent association in the adjusted model, the elevated unadjusted odds point to a possible vulnerability among immigrant and ethnic minority populations. These groups may face structural, linguistic, or cultural barriers that affect treatment adherence, and future interventions should consider these dimensions to enhance equity in LTBI care.

To the best of our knowledge, this study is the first in Malaysia to examine the proportion of incomplete LTBI preventive treatment and its associated factors at the subnational level. The retrospective cohort study design using secondary data was cost-effective and less time-consuming than a prospective cohort study. However, there are a few limitations that should be acknowledged. Firstly, secondary data was subjected to missing data. The 188 missing ethnicity data exemplified this. Additionally, the study was constrained by the limited scope of variables available in the LTBIS 401A registry. Important sociodemographic and behavioral factors—such as education level, socioeconomic status, and lifestyle behaviors including tobacco and alcohol use—were not captured and therefore could not be analyzed. Furthermore, the registry lacked data on nutritional status (*e.g.*, body mass index), which is another important determinant of treatment outcomes. Poor nutrition can reduce drug tolerance and impair immune function, increasing the risk of adverse effects and treatment discontinuation. In this study, all 24 individuals who experienced adverse drug reactions (ADRs) failed to complete LTBI preventive treatment, underscoring the significant role of side effects in non-completion. Although ADRs were recorded, detailed clinical management information was not available. The absence of nutritional data further limited the ability to assess this factor.

Taken together, these limitations highlight the need to enhance LTBI surveillance systems by incorporating a broader range of behavioral, socioeconomic, and clinical variables. Future research should include nutritional assessments and more detailed ADR monitoring to better understand their impact on treatment adherence and outcomes. Such improvements would strengthen program evaluation and support more targeted and effective public health interventions. Several recommendations can be made to improve the completion rate of LTBI preventive treatment in Sabah, Malaysia. Non-HCW, residents of the Tawau Division, and individuals with LTBI without contact with TB patients were identified as specific groups requiring tailored interventions to increase the completion rates. Targeted interventions can help to improve the outcome of LTBI preventive treatment and improve the overall effectiveness of TB control efforts in Sabah, Malaysia. Furthermore, as TB is a social disease, tackling TB requires the coordinated action of many partners working together across government agencies, NGOs, and community boundaries. Further research is needed to understand better the reasons for incomplete LTBI preventive treatment, particularly among individuals with LTBI who are non-HCW, residents of Tawau Division, and non-contact to TB patients. This could involve qualitative studies to explore patient attitudes and beliefs towards LTBI and preventive treatment and identify barriers to completion.

In addition, we recommend expanding LTBI surveillance systems such as the LTBIS 401A registry to include key individual-level variables such as nutritional status (*e.g.*, body mass index) and behavioral risk factors, including tobacco and alcohol use. This will

facilitate more comprehensive risk profiling and inform the development of evidence-based, patient-centered interventions.

## CONCLUSIONS

The proportion of incomplete preventive treatment among individuals with LTBI in Sabah was comparatively lower than many previous studies. Targeted interventions should be developed to address the specific needs of the groups with higher odds of having incomplete preventive treatment. It includes tackling the social determinants of health, like improving healthcare system accessibility. A prospective study to evaluate these interventions' effectiveness in improving preventive treatment completion rate is recommended.

## ACKNOWLEDGEMENTS

We would like to acknowledge the Sabah State Health Department for their logistical support, which was instrumental in facilitating smooth data acquisition. During the preparation of this work, the authors used ChatGPT 3.5 in order to correct English. After using this tool, the authors reviewed and edited the content as needed and take full responsibility for the content of the publication.

### Funding
The research was funded by the Universiti Sains Malaysia through Tabung Insentif Pembangunan Pengajian Siswazah (TIPPS). The funders had no role in study design, data collection and analysis, decision to publish, or preparation of the manuscript.

### Grant Disclosures
The following grant information was disclosed by the authors:
Universiti Sains Malaysia.

### Competing Interests
The authors declare there are no competing interests.

### Author Contributions
- Ahmad Firdaus Mohamed conceived and designed the experiments, performed the experiments, analyzed the data, prepared figures and/or tables, authored or reviewed drafts of the article, and approved the final draft.
- Mohd Nazri Shafei conceived and designed the experiments, performed the experiments, analyzed the data, prepared figures and/or tables, authored or reviewed drafts of the article, and approved the final draft.
- Wan Mohd Zahiruddin Wan Mohammad performed the experiments, analyzed the data, prepared figures and/or tables, authored or reviewed drafts of the article, and approved the final draft.

- Roddy Teo conceived and designed the experiments, authored or reviewed drafts of the article, and approved the final draft.
- Amabel Seow Min Hui performed the experiments, authored or reviewed drafts of the article, and approved the final draft.

## Human Ethics

The following information was supplied relating to ethical approvals (i.e., approving body and any reference numbers):

The study was approved by the Malaysian Medical Research and Ethics Committee (NMRR ID-22-02792-6PV (IIR)) and the Human Research Ethics Committee of Universiti Sains Malaysia (USM/JEPeM/22110728). To maintain data confidentiality, participant identities were anonymized, and all data were securely stored with limited access granted solely to authorized personnel. Subsequently, all remaining data was coded to provide additional protection for participant confidentiality.

## Data Availability

The raw measurements are available in the Supplementary Files 1.

## Supplemental Information

Supplemental information for this article can be found online at http://dx.doi.org/10.7717/peerj.19736#supplemental-information.

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
