# Peer review of "Factors associated with incomplete latent tuberculosis infection preventive treatment in Sabah, Malaysia"

_PeerJ, doi:10.7717/peerj.19736_

## Round 0.1 · original submission · Major Revisions

Please address all authors comments.

Reviewer 1 ·

Basic reporting

Congratulations to the authors for their research. The paper offers an interesting perspective on latent tuberculosis research. It is an easy-to-read, clear, and coherent manuscript. The bibliography is sufficient.

Experimental design

In the context of globalization and population migration, the study brings a necessary perspective to the health policies that states must implement in a unified manner to limit the spread of tuberculosis. The manuscript complies with the journal's criteria.

Validity of the findings

The discussions are extensive and well presented, and the conclusions underline the importance of the research topic.

·

Basic reporting

Table 1: Comparison word can be removed as no statistical comparison is provided in this table

Experimental design

-

Validity of the findings

-

Additional comments

This study is conducted in Sabah, Malaysia, to identify the proportion of individuals with LTBI and further to assess the factors associated with the completion/ non-completion of LTBI treatment. I would like to commend the efforts taken by the authors for conducting this study, which is much needed for the implementation of LTBI treatment and also policy changes as a long-term vision. However, there are a few comments from my end, which are mentioned below.

Entire manuscript: The individuals with LTBI are referred to as patients. I am aware that the guidelines might mention individuals with LTBI as patients, but this is more stigmatizing. We are well aware that TB infection and TB disease are two entirely different entities, and labelling them as patients wouldn't be well suited.

Materials and methods:
1. Research tool: A few important variables that are related to treatment completion are not captured here. Consumption of tobacco/ alcohol is not captured, which is a known risk factor (https://pubmed.ncbi.nlm.nih.gov/28720633/). The nutritional status of the individual is not captured. Simple BMI would help in the assessment of nutritional status.

Results:
1. Mention that none of the individuals on LTBI treatment had CRF, cancers, on steroid treatment, etc
2. Two individuals had an HIV reactive status, and one had incomplete treatment. Was any significant factor that led to the incomplete treatment in this individual noted?

Discussion:
1. Nutritional status and adverse drug reactions are not considered as possibilities when the factors among non-HCWs are discussed.
2. 3% of the health care workers still had incomplete treatment, which can be due to various reasons. (https://pubmed.ncbi.nlm.nih.gov/34184972/) Further discussion on these topics is needed.
3. Nationality and ethnicity have not been discussed here at all. Both variables have a significant association with treatment outcomes. Immigrants and individuals from ethnicities other than Sabahan had higher proportions of non-completion.

Discussion in general needs to be strengthened.

Recommendations:
Need to add recommendations for a comprehensive collection of individual details, including nutritional assessment and substance abuse (tobacco consumption).

·

Basic reporting

The article is technically sound. Good introduction and well-explained background information.

Experimental design

The research question is well defined, relevant, and meaningful.

Validity of the findings

To draw a conclusion and recommendation, more understanding is required of the process and mechanism of the implementation strategies of TPT. Although the overall incomplete treatment was 9.2%, the author did not explain the strategies (drug supply, DOT, etc.) of implementing TPT in different settings (non-HCW occupation residents of Tawau Division, and LTBI patients without contact to TB patients).

The reviewer suggests explaining the TPT implementation strategies.

---

## Round 0.2 · accepted · Accept

Thanks for addressing the reviewers' comments!

·

Basic reporting

No comment

Experimental design

No comment

Validity of the findings

No comment

Additional comments

The authors have addressed the queries raised by me in the previous version of the manuscript.

·

Basic reporting

The remarks made earlier on the TPT implementation strategies have been addressed.

Experimental design

explained during the1st review.

Validity of the findings

no further remarks